# Neurophysiological Changes Induced by Music-Supported Therapy for Recovering Upper Extremity Function after Stroke: A Case Series

**DOI:** 10.3390/brainsci11050666

**Published:** 2021-05-20

**Authors:** Shashank Ghai, Fabien Dal Maso, Tatiana Ogourtsova, Alba-Xifra Porxas, Myriam Villeneuve, Virginia Penhune, Marie-Hélène Boudrias, Sylvain Baillet, Anouk Lamontagne

**Affiliations:** 1School of Physical and Occupational Therapy, McGill University, Montreal, QC H3G 1Y5, Canada; tatiana.ogourtsova@mail.mcgill.ca (T.O.); mh.boudrias@mcgill.ca (M.-H.B.); anouk.lamontagne@mcgill.ca (A.L.); 2Feil & Oberfeld Research Centre of the Jewish Rehabilitation Hospital–CISSS Laval, A Research Site of the Centre for Interdisciplinary Research of Greater Montreal (CRIR), Laval, QC H7V 1R2, Canada; myriam.villeneuve@mail.mcgill.ca; 3Laboratory of Simulation and Movement Modelling, School of Kinesiology and Physical Activity, Université de Montréal, Montreal, QC H3T 1J4, Canada; fabien.dal.maso@umontreal.ca; 4Centre Interdisciplinaire de Recherche sur le Cerveau et l’Apprentissage, Montréal, QC H7N 0A5, Canada; 5Graduate Program in Biological and Biomedical Engineering, McGill University, Montreal, QC H3A 0C3, Canada; alba.xifraporxas@mail.mcgill.ca; 6Department of Psychology, Concordia University, Montreal, QC H3G 1M8, Canada; Virginia.Penhune@concordia.ca; 7Laboratory for Brain Music and Sound (BRAMS), Centre for Research in Brain, Language and Music, Montreal, QC H2V 2S9, Canada; 8McConnell Brain Imaging Centre, Montreal Neurological Institute, McGill University, Montreal, QC H3A 2B4, Canada; sylvain.baillet@mcgill.ca

**Keywords:** neurorehabilitation, motor learning, functional connectivity, auditory-motor coupling

## Abstract

Music-supported therapy (MST) follows the best practice principles of stroke rehabilitation and has been proven to instigate meaningful enhancements in motor recovery post-stroke. The existing literature has established that the efficacy and specificity of MST relies on the reinforcement of auditory-motor functional connectivity in related brain networks. However, to date, no study has attempted to evaluate the underlying cortical network nodes that are key to the efficacy of MST post-stroke. In this case series, we evaluated changes in connectivity within the auditory-motor network and changes in upper extremity function following a 3-week intensive piano training in two stroke survivors presenting different levels of motor impairment. Connectivity was assessed pre- and post-training in the α- and the β-bands within the auditory-motor network using magnetoencephalography while participants were passively listening to a standardized melody. Changes in manual dexterity, grip strength, movement coordination, and use of the upper extremity were also documented in both stroke survivors. After training, an increase in the clinical measures was accompanied by enhancements in connectivity between the auditory and motor network nodes for both the α- and the β-bands, especially in the affected hemisphere. These neurophysiological changes associated with the positive effects of post-stroke MST on motor outcomes delineate a path for a larger scale clinical trial.

## 1. Introduction

Stroke is a leading cause of disability that can result in a contralesional upper extremity paresis [1,2], including impaired gross and fine motor functions, changes in muscle tone, and reduced range of motion [3,4]. Current rehabilitation approaches often yield modest to moderate motor improvements [5,6,7], with residual upper extremity impairments becoming permanent and leading to activity restrictions [8] and reduced quality of life [2,9]. Longitudinal studies have shown that 46% to 66% of stroke survivors do not regain functional independence in the upper extremity 6 months post-stroke [3,4,10,11]. Best practice principles in stroke rehabilitation indicate that interventions should be individually tailored, meaningful, task specific, variable, and should involve sufficient repetition and challenge to promote recovery [12,13,14,15]. Previous studies including the ones from our laboratory [16,17], indicate that music-supported therapy (MST) can not only meet but extend beyond these imperatives to yield improvements in motor skills, cognitive functions [18,19,20,21,22,23,24,25,26,27], and stress reduction [28]. 

The efficacy and specificity of MST are hypothesized to be mediated by the auditory-motor network, which is required to play music as well as to support recovery and/or compensate for stroke-related dysfunction [29,30]. Pascual-Leone’s work clearly demonstrates that training with a musical instrument such as piano can instigate neural plasticity by inducing swift unmasking of existing synapses and the formation of newer ones [31]. In expert and novice musicians, MST-induced auditory-motor coupling engages a network of distributed brain regions that includes the auditory and primary motor cortices (M1s), the dorsal and ventral parts of the premotor area (PMd and PMv), the inferior frontal gyrus (IFG), and the supplementary motor area (SMA) [29,32]. In stroke survivors, MST has been shown to increase motor cortex excitability in the affected hemisphere and to be associated with partial recovery of motor functions of the paretic hand [17,18,33]. Recent systematic reviews and meta-analyses [18,19,34] have also highlighted the beneficial effects of MST on upper extremity recovery in chronic stroke survivors. In the context of this study, we sought to clarify the neurophysiological mechanisms that underlie the beneficial effects of MST when used as a tool for rehabilitation post-stroke [18,19].

Neuroimaging techniques have been used to evaluate the neurophysiological effects associated with MST in stroke survivors [33,35,36,37] during passive listening [36,37] or silent tapping of musical instruments [33]. Using functional magnetic resonance imaging (fMRI), Rojo et al. [37] reported greater activation in motor areas contralateral to the affected upper limb during passive music listening after MST. They also observed bilateral enhanced cortical excitability as indexed by a larger amplitude of motor evoked potentials (MEP) that were evoked with transcranial magnetic stimulation (TMS). Amengual et al. [35] also reported MEPs’ enhancement following MST, although it was restricted to the lesioned hemisphere. Using magnetoencephalography (MEG), Fujioka et al. [36] observed event-related desynchronization (ERD) (i.e., power decrease) in the β-band (15–35 Hz) in auditory and sensorimotor cortices after MST, while stroke participants were passively listening to a metronome. The focus on neurophysiological effects in the α- and β-bands is due to their remarkable signal strength in humans, and their well-studied association with cognitive vigilance [38,39] and motor performance [30], respectively. A critical electroencephalography (EEG) study by Altenmuller et al. [33] also evaluated ERD and intracortical connectivity changes in the α- and β-bands in stroke participants actively playing a muted-drum or a muted-piano instrument. The authors reported greater ERD in the β-band during silent playing after MST, and no differences in the α-band. They also observed increased β-band intra- and interhemispheric coherence between the frontal and parietal regions when stroke participants played the muted electronic drum sets using either the affected or unaffected arms after MST. Here too, no pre/post MST changes in coherence were observed by the authors in the α-band. The authors interpreted these effects as reflecting increased auditory-motor coupling in stroke survivors following MST. From the current state of literature, gaps in knowledge exist regarding the extent of changes in the functional connectivity between the auditory-motor network nodes after MST. Moreover, the literature fails to explain as to how MST-induced functional connectivity changes might differ in stroke survivors not presenting the same level of motor impairment at baseline.

MST studies involving stroke participants have reported treatment effects in pre-determined regions of interest (ROIs) of the auditory-motor network, such as in M1 and the auditory cortex (AC). However, because stroke causes changes in functional brain organization, ROIs based on standard coordinates or atlases of brain anatomy may not be accurate in individuals with brain lesions within these areas [40,41,42,43]. We therefore sought to determine how the neurophysiological effects of MST are related to clinical outcomes, using brain ROIs selected based on functional localizers in the motor and sensory regions in chronic stroke participants. We anticipated that MST would induce enhanced functional coupling between multiple nodes of the auditory-motor network following an intensive, 3-week piano training intervention in stroke survivors. Furthermore, previous data from our laboratory showed that stroke survivors who had greater functional status prior to MST experienced the largest gains in manual dexterity and functional use of their upper extremity following MST [17]. In the present study, we therefore hypothesized that baseline motor performance would be predictive of changes in functional connectivity in the auditory-motor network following MST. We anticipate observing larger post-MST enhancements in auditory-motor functional connectivity in stroke survivors with better baseline functional capability as compared to stroke survivors with poorer baseline capability.

## 2. Materials and Methods

### 2.1. Participants

Seven chronic stroke survivors were enrolled from the discharged list of a rehabilitation center in the greater Montreal area. However, five of these participants had to be excluded because of disruption in their MST due to COVID-19 pandemic restrictions. Two participants (Table 1) were included based on their chronicity (6–24 months post-stroke), site of lesion (middle cerebral artery), and residual capacity to dissociate active wrist and finger movement in the affected paretic extremity (i.e., a score of 3 to 6 out of 7 on the arm and hand components of the Chedoke McMaster Stroke Assessment [44]). Moreover, participants had to have corrected-to-normal visual and auditory acuity. Those with moderate to severe cognitive deficits (scores ≤ 23 on the Montreal Cognitive Assessment [45]) and visuospatial neglect (6 omissions or more on the Bell’s test [46]) were excluded. Individuals were also excluded if they were still receiving therapy for the upper extremity or if they had another condition interfering with upper extremity movements. Moreover, individuals with prior professional experience of music, i.e., more than 1 h per week of practice of any musical instrument during the past 10 years, were not included in the study. The study was approved by the Institutional Research Ethics Board and written informed consent was obtained from each participant. 

### 2.2. Procedures

This study involved a multiple pre- multiple post-sequential design, with participants assessed at baseline (week 0), pre-MST intervention (week 3), post-MST intervention (week 6), and at follow-up (week 9). The MST intervention spanned over 3 consecutive weeks and involved three individual 1-h sessions of supervised piano lessons per week, for a total of nine sessions. During the supervised training sessions, participants played on a touch-sensitive Yamaha P-155™ piano keyboard. They received auditory feedback on their performance through Synthesia™ and verbal feedback on the quality of movements and compensatory strategies by the therapist. The supervised sessions were further complemented by a biweekly home-based training program of 30 min per session. This portion of the training was performed independently by the participants on a roll-up flexible piano, without Synthesia™. The patients also kept a log of their home practice, which was reviewed by the training clinician at every visit. Clinical tests were performed at every assessment time point (week 0, week 3, week 6, and week 9) and included an assessment of gross (Box and Block test (BBT) [47]) and fine motor skills (nine-hole peg test (NHPT) [48]), arm movement coordination (finger-to-nose test [49]), finger movement coordination (finger-tapping test [50]), functional use of the upper extremity (Jebsen hand function test (JHFT), average of subset 2 to 7 [51]), and grip strength (JAMAR hand-held dynamometer [52]). A similar protocol has been described in an earlier publication from our laboratory [17]. 

#### 2.2.1. MEG Acquisition and Protocol

MEG recordings were acquired pre-training (1 day before the first MST session) and post-training (1 day after the last MST session) using a CTF, 275-channel system. 3D digitization of individual head shapes was obtained with a Polhemus Fastrak, using approximately 100 uniformly distributed head points. MEG signals were sampled at 2.4 kHz. The individual anatomical brain volume was obtained for each participant on a 1.5T MRI scanner (Siemens Prisma; TR = 2.3 s, TE = 2.32 s, TI = 0.9 s, flip angle = 8.00; 1 × 1 × 1 mm^3^ voxels) immediately following the last MEG session. Head localization coils (nasion, left, and right pre-auricular) and the head-surface points were used for co-registration between MEG and MRI coordinate systems. We also collected an empty room recording of 5 min, before both MEG sessions, for noise modeling in MEG analyses. 

#### 2.2.2. MEG Acquisition

The MEG session consisted of the following runs, with the order of runs 2 to 4 randomized between participants: (Block 1) One 5-min run with participants asked to remain still and awake (resting-state), with their eyes open fixating a crosshair on an empty black screen; (Block 2) one run with participants exerting submaximal dynamic and isometric force handgrips (Current Designs Inc, Philadelphia, PA, USA) (50 repetitions of 3 s interspersed by 5 s of rest, 15% maximum voluntary contraction); (Block 3) one run with participants playing a piano melody over 3 to 5 min—actual duration was dependent upon the time taken by the participant to play the entire musical piece. Participants used a MEG-compatible piano keyboard designed by Hollinger et al. [53] and were instructed to play a simple standardized musical sequence with the use of visual display from Synthesia™; (Block 4) one run during which participant listened to the standardized melody they were expected to play (auditory piece duration: 144 s) and; (Block 5) a final run during which participants were played back and passively listened to the melody they themselves played in Block 3.

#### 2.2.3. MEG Data Analysis

All MEG data analyses were performed using the open-source software Brainstorm [54]. The MEG data were filtered using a notch filter at 60, 120, and 180 Hz, bandpass filtered from 1 to 150 Hz, and subsequently down sampled to 120 Hz. Artifacts due to heartbeats and blinks were removed using a signal space projection and independent component analysis. All single MEG trials were inspected visually and trials with artifacts were removed from further analysis. Since the focus of the present study was passive listening condition, we imported 142 s of MEG recordings during which the stroke participants listened to the standardized melody clip (Block 3). 

The MEG forward fields were obtained using an overlapping spheres head model that approximates head shape with a set of locally fitted spheres under each sensor [55]. Source modeling used the weighted-minimum norm imaging method with Brainstorm’s default parameters [56]. The dipole orientations were kept constant as they were constrained perpendicularly with respect to the cortical surface. Moreover, we used depth weighting with an order of 0.5 and maximal amount of 10. We used the empty-room recordings to derive session-specific empirical sensor noise covariance sample statistics to inform source modeling. Cortical currents were sampled using 15,000 elementary current dipoles distributed over the cortical surface, oriented perpendicularly to the local cortical surface. We obtained power spectrum density estimates of ROI source time series in both participants using Welch’s method (1-s time windows with 50% overlap: Figure 1). We also developed the power spectrum density maps for the α- (8–12 Hz) and the β- (12–38 Hz) bands for indicating the relative power of the cortical signals (Appendix A). We then estimated frequency-specific coherence to assess functional connectivity between the ROIs in the α- and the β-bands. Coherence is a Fourier-based signal technique that assesses cross-spectral interactions in phase (phase-locking) and amplitude (amplitude envelope correlation) between a pair of signals at each frequency bin on the spectrum [54]. 

#### 2.2.4. Regions of Interest

We selected six ROIs from their reported involvement in auditory-motor functions (Figure 2) [29,57]. These included, in both hemispheres: M1, PMd, PMv, SMA, IFG, and the AC. SMA, PMv, and PMd were defined according to Boudrias et al. [58]. IFG was defined from an atlas of Broadmann areas [59]. M1 was localized bilaterally for each patient as the set of cortical MEG sources with the strongest (top 5%) ERD, i.e., a marked decrease of β power over central regions during Block 2’s isometric handgrips. We measured ERD over the three seconds of handgrip as participants were maintaining 15% of their maximum voluntary contraction [60]. We functionally localized AC bilaterally from brain activity during the passive listening condition (Block 3): we averaged MEG single-trial epochs around each piano key press, and defined AC as the cluster of cortical sources with the strongest peak activity at about 90 ms [61,62] following the key press. We localized M1 and AC pre- and post-MST. The bilateral AC locations were similar before/after MST in both participants. We noted a shift in M1 location following MST in both participants.

## 3. Results

### 3.1. Clinical Assessment

The clinical assessment scores of both participants at the different time points are provided in Table 2. We used the average scores at baseline and the pre-MST blocks (week 0, week 3) to compare the performance of the participants after the training (week 6) and during the follow-up period (week 9). Both participants improved on all clinical outcome measures at post-intervention (range of improvement: 8.7–62.6%), with the exception of NHPT for Participant 1 (a −4% improvement). Performance improvements sustained or increased at follow-up (week 9), especially for Participant 2, in BBT, NHPT, and JHFT. Overall, the clinical performance of Participant 2 showed larger percentage increases after MST as compared to Participant 1.

### 3.2. Functional Connectivity

#### 3.2.1. α-Band

In Participant 1, there was increased coherence between most region pairs of the unaffected (left) hemisphere post-MST (40% to 767%; Figure 3). In the affected (right) hemisphere, we observed a general reduction of coherence between most ROIs (−3% to −75%), with the exception of PMv-AC, which showed a 47% increase post-training.

Data from Participant 2’s left hemisphere showed increased α-band coherence ranging from 12.1% to 104.3% between AC and all tested sensorimotor ROIs. The increase in coherence between PMd-AC was the smallest (3.8%). We observed similar effects post-MST in the right (affected) hemisphere, with α-band coherence increases ranging from +53.1% to +262.2% between AC and M1, SMA, PMv, PMd and IFG. 

#### 3.2.2. β-Band

Cortico-cortical coherence values in the β-band during the passive listening condition are shown in Figure 4. In Participant 1, the unaffected (left) hemisphere showed a general reduction following MST between most of the sensorimotor ROIs (−17% to −73%), except SMA-AC. In the affected (right) hemisphere, there was increased coherence between most sensory and motor regions (17% to 29%), with the exception of SMA-AC (−30%) and PMd-AC (−44%). 

Participant 2 showed a general reduction (−7.5% to −53.8%) post-training in β-band coherence between most sensorimotor brain regions in the unaffected (left) hemisphere and the AC (i.e., SMA-AC, PMv-AC PMd-AC, and IFG-AC), except for left M1-AC, which showed a 19.3% increment. In the affected (right) hemisphere, all ROIs showed increased (98.6–394.7%) β-band coherence with AC, with the exception of PMv-AC (+2%).

## 4. Discussion

We reported a case-series study of pre- and post-MST changes in α- and β-band cortico-cortical coherence as a measure of functional connectivity between nodes of the auditory-motor network, following a 3-week MST in two chronic stroke survivors. For the first time, we observed changes in functional coherence between different nodes that included an increase in connectivity between the auditory and motor cortex in the affected hemisphere, due to MST training. Such neurophysiological changes were accompanied by enhancement of manual dexterity and upper extremity use in both participants immediately after MST and at the 3-week follow-up. We also observed that the magnitude of changes in the functional coherence were different in both the participants, which is likely due to the differences in the pre-MST clinical profile. 

### 4.1. Behavioral Observations 

Improved clinical scores of the NHPT, JHFT, BBT, and finger-tapping test were observed for both participants after MST, particularly for Participant 2. This observation may be in part accountable for differences in the physical capabilities observed between the participants at baseline (see Table 2). Indeed, Participant 1’s post-training performances approached closer to age-related normative values for the JHPT, NHPT, BBT, and finger-tapping test (Table 2), hence reaching a possible ceiling effect. Comparatively, Participant 2’s post-MST clinical performances remarkably improved in tasks assessing hand/arm function, grip strength, coordination, and dexterity. We emphasize that prior to the MST intervention, Participant 2 was not able to produce fine motor tasks, as measured by the NHPT. 

We also noted that both participants showed improved gross manual dexterity and upper extremity use in functional activities (Appendix A). These results align with previous work by Lang et al. [68], Ranganathan et al. [69], and Villeneuve et al. [17] who also demonstrated that improvement of finger motor performances translated to hand motor performances. 

### 4.2. Neurophysiological Observations

In addition to clinical improvements, we report post-MST neurophysiological changes in functional cortico-cortical connectivity, primarily in the form of increased α- and β-band coherence between nodes within the auditory-motor network, especially in the affected hemisphere. Previous studies have reported that an enhancement of connectivity in the same frequency bands in the affected hemisphere after a therapeutic intervention, such as physical therapy intervention or brain computer interface training, is associated with functional recovery in the chronic, post-stroke phase [70,71,72]. Arce-McShane et al. [73] further claimed that increased coherence between sensorimotor areas during a force production task could be a vector of neural signal enhancement of sensorimotor integration during motor task performance. Along the same line, Nicolo et al. [71] proposed that this effect could be associated with the plastic reorganization of feedforward and feedback neurophysiological signaling of internal models of behaviour, and considered it a possible pathway for novel interventions in stroke rehabilitation. They argued that enhanced β-band connectivity between motor areas could be a marker of the formation and strengthening of synaptic connectivity in stroke survivors. Similarly, Dubovik et al. observed that an increased α-band functional coherence between the cortical network nodes, such as the motor and pre-motor cortex post-stroke, was linearly correlated with enhancements of motor and cognitive functions [74]. The authors also mentioned that the phase locking in the α-band could represent the enhanced behavioral performance of a task by improving long-term potentiation, conscious perception, and perceptual discrimination. In our present case series, the increased functional coherence in the α- and β-frequency bands could therefore be inferred as a plastic reorganization of the brain structures that led to enhancements in clinical outcomes.

Our data showed enhanced β-band coherence between M1 and AC in the affected (right) hemisphere of both participants. This suggests that enhanced connectivity is present while listening passively to a trained musical piece, which could be the result of a more synchronized activity developed after the musical entrainment of the auditory-motor network [75,76,77]. We also observed enhanced coherence between SMA and IFG in the affected hemisphere of Participant 2, but not in Participant 1, the former being more severely affected in terms of upper extremity function compared to the latter. A possible reason for such differential effects between participants could be due to the different roles played by IFG and SMA’s when sequencing and predicting motor events, such as during musical tasks. These regions are also involved in hand motor control [78,79]. Therefore, the observed interindividual variability in coherence may reflect the large differences observed at baseline in terms of fine and gross motor performances between subjects. 

We also observed a large enhanced PMd-AC coherence in the affected hemisphere of Participant 2. Moreover, we observed enhanced values in terms of α-band connectivity between PMv-AC for both participants. No notable changes were detected in β-band PMv-AC connectivity in Participant 2. These differences in post-MST connectivity changes involving PMv-AC and PMd-AC may be related to the respective roles of these structures in hand motor performances. From animal studies, PMv is key to posture and the recruitment of intrinsic muscles of the hands for grasping objects, while PMd is associated with controlling proximal forearm muscles during hand lifting, i.e., grip-lift synergy while performing sequential movements [80]. PMd is also more activated than PMv during finger tapping [80,81]. Such prior evidence may explain why our data showed that Participant 2 performed better in the finger-tapping test post-MST intervention, compared to Participant 1. 

We found different coherence patterns between nodes of the non-affected hemisphere in the participants according to their pre-MST clinical scores. In Participant 2, we observed enhanced α-band connectivity post-MST between most nodes, with the exception of PMd-AC, where enhancement was minimal. Similar enhancements in connectivity were also observed in the β-band; however, these were restricted to interactions between M1-AC; β-band coherence between all other network nodes was reduced post-MST. In Participant 1, we observed an overall α-band coherence increase between all network nodes tested. β-band coherence reductions were also observed in the non-affected hemisphere, except for SMA-AC. Moreover, we also observed that the magnitude of change in the coherence after MST was larger for Participant 2 as compared to Participant 1 in both the frequency bands. These observations align well with two standing mechanisms in the literature. In the first mechanism, studies have reported post-stroke enhanced activation in the non-affected hemisphere, especially in M1, with respect to a proportional inhibition of the affected M1 [82,83,84]. Therefore, any reduction in the involvement of the non-affected hemisphere could be considered as an improvement in the functional reorganization of the affected M1. In Participant 1, a reduction in coherence measures was observed in the non-affected hemispheric together with an increase in coherence in the affected hemisphere after MST, which supports this view. The second mechanism stipulates that a functional reorganization of the non-affected hemisphere in severe stroke supports the recovery of paretic limb, based on observations of associations between enhanced activation in the non-affected hemisphere and greater motor outcome in stroke survivors [82,85,86]. In Participant 2, the increased connectivity observed in the non-affected hemisphere could be related to the functional reorganization of the non-lesioned hemisphere that led to improved control of the paretic upper extremity in stroke survivors [82,83,84]. For instance, Riecker et al. [82] have reported that enhanced activation of the non-lesioned hemisphere in severe stroke might represent additional recruitment of the neural structures that compensates for increased demands on the damaged lesioned motor system. Participant 1, however, was closer to ceiling performances at baseline with respect to age-related standards and was less challenged by the training, which may have required lesser recruitment of his non-affected hemisphere. Based on such observations, it can be interpreted that the initial pre-training profile of a participant matters and that the magnitude and nature of changes in coherence after MST could be based on the pre-MST clinical levels of a participant. However, this would need to be confirmed in a larger sample, as a causal relationship cannot be established based on two cases.

To our knowledge, only one study has reported post-MST connectivity changes in stroke survivors [33]. In this study, an overall increased coherence between all EEG electrodes was found after a 3-week MST. However, they did not include electrodes above the temporal lobes in their analysis, which challenges the interpretability of the scalp data in terms of auditory-motor coupling. 

### 4.3. Limitations 

A number of limitations were present in this case report. Firstly, both participants had a stroke localized in their right hemisphere, which limits the generalizability of the neurophysiological effects reported in the affected vs. non-affected hemispheres, respectively. We aim to expand the present findings to a larger clinical trial, where the number of left and right hemispheric stroke participants will be balanced. Secondly, we used functional localizers to map M1 and AC individually, but we used templates to define the other ROIs, which yielded approximate locations with respect to individual anatomy. Thirdly, we used a gripping task as a functional localizer of M1, instead of a finger-tapping test, which was the movement trained in the MST piano task. This could have biased our interpretation of M1-related effects from a region defined more broadly than the swath of cortical regions primarily involved in fine finger movements. Fourthly, this study used a single subject ‘multiple pre- multiple post-sequential design’ instead of using a placebo group. While such a design does not allow for concluding on comparative effectiveness (or to dissociate the placebo effect from actual intervention effect), it is appropriate given the purpose of the study, which was to examine the neurophysiological changes induced by MST and how these changes are related to the clinical profile of the participants.

## 5. Conclusions

This case series is indicative of motor and neurophysiological changes associated with a 3-week MST in chronic stroke survivors. Overall, we observed increased α- and β-band coherence within the auditory-motor network after MST. This study provides early evidence of the mechanistic foundations and neurophysiological markers of MST-induced improvement of upper extremity functions after stroke. We look forward to these findings contributing to delineating a path for future clinical trials in our laboratory. Additionally, in future trials we intend to evaluate the global mapping of possible brain regions showing variations of CMC with motor functions. These changes will help us identify which brain nodes in motor circuits interact with the effectors (hand muscles) to support the recovery of paretic hand after MST.

## Figures and Tables

**Figure 1 brainsci-11-00666-f001:**
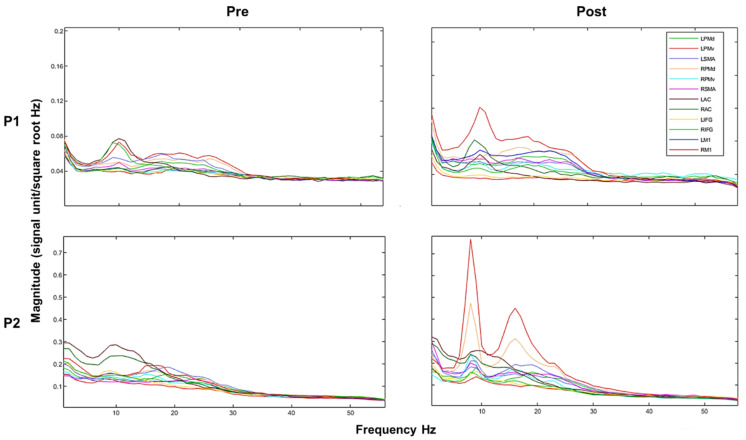
Power spectrum density plots for both participants, pre- and post-MST. (P1: Participant 1, P2: Participant 2, L: left, R: right, Pre: pre-training; Post: post-training, M1: primary motor cortex, AC: auditory cortex, SMA: supplementary motor area: PMv: premotor ventral area, PMd: premotor dorsal area, IFG: inferior frontal gyrus).

**Figure 2 brainsci-11-00666-f002:**
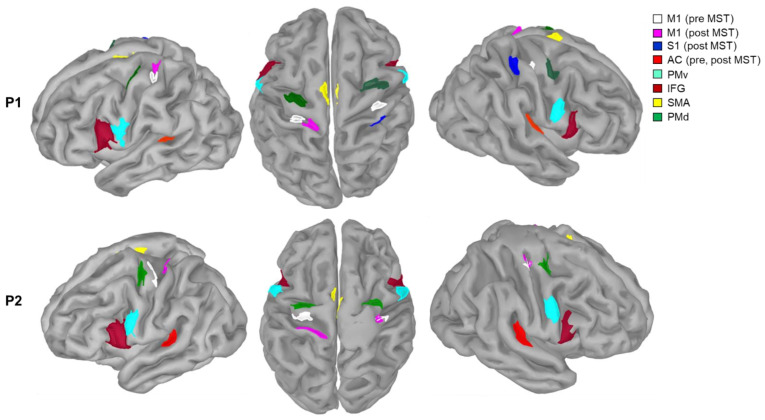
Regions of interests of the auditory-motor network created by Brainstorm. M1 and AC ROIs were defined from individual functional data pre- and post-MST; SMA, PMd, and PMv were selected from a previously published study [58]; IFG was selected from the Broadmann area maps’ atlas [59] (P1: Participant 1, P2: Participant 2, M1: primary motor cortex, S1: somatosensory cortex, AC: auditory cortex, IFG: inferior frontal gyrus, pre-MST: before music-supported therapy, post-MST: after music-supported therapy, SMA: supplementary motor area: PMv: premotor ventral area, PMd: premotor dorsal area).

**Figure 3 brainsci-11-00666-f003:**
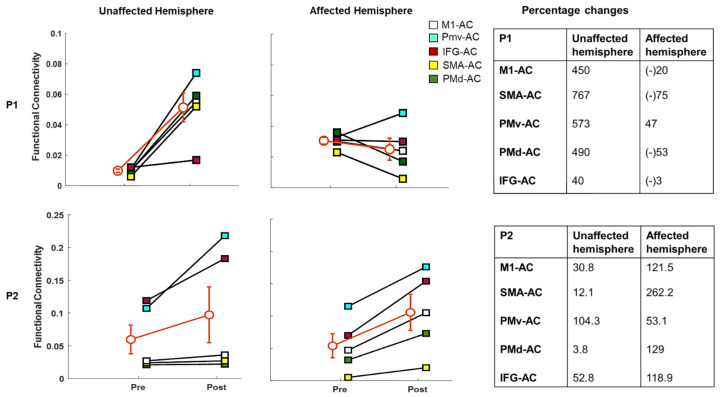
Intracortical coherence in the α-band in Participant 1 and Participant 2, with percentage connectivity changes before and after MST. The error bars (i.e., orange line) represents the standard error of the mean in the overall functional connectivity. (M1: primary motor cortex, AC: auditory cortex, SMA: supplementary motor area: PMv: premotor ventral area, PMd: premotor dorsal area, IFG: inferior frontal gyrus, P1: Participant 1, P2: Participant 2).

**Figure 4 brainsci-11-00666-f004:**
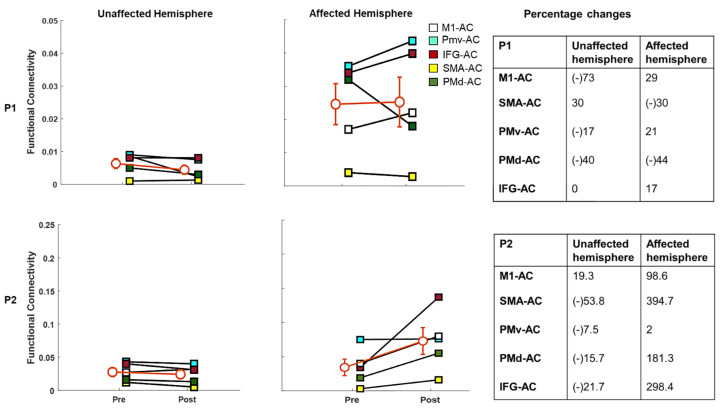
Intracortical coherence in the β-band in Participant 1 and Participant 2 with percentage connectivity changes before and after MST. The error bars (i.e., orange line) represent the standard error of the mean in the overall connectivity. (M1: primary motor cortex, AC: auditory cortex, SMA: supplementary motor area: PMv: premotor ventral area, PMd: premotor dorsal area, IFG: inferior frontal gyrus, P1: Participant 1, P2: Participant 2).

**Table 1 brainsci-11-00666-t001:** Characteristics of stroke participants.

	Participant 1	Participant 2
Age (years)	66	67
Gender (M/F)	M	M
Affected hemisphere	Right	Right
Nature of CVA	Ischemic	Ischemic
Time after stroke (months)	15	15
Site of lesion	Subcortical	Lacunar internal capsule
CMSA arm/hand score	5/5	3/3
Piano experience (years)	0	0
Handedness	Right	Right

CMSA: Chedoke–McMaster Stroke Assessment, CVA: Cerebrovascular accident.

**Table 2 brainsci-11-00666-t002:** Descriptive data from clinical assessments.

	P1						P2						Age Norms **
	Base	Pre	Post	%	F-up	%	Base	Pre	Post	%	F-up	%	
Nine-Hole Peg Test (s)	44	47.2	43.7	(−)4	45.9	1	120 +	120 +	109.6	(−)9	102.3	(−)15	21.6 ± 2.9 [63]
Box and Block Test (n)	41	42.6	46.3	11	45.6	9	19.6	23.6	24.6	14	29.3	35	67.4 ± 7.8 [64]
Jebsen Hand Function Test, subset 2–7 (s)	9.2	8.4	7.8	(−)28	7.1	(−)35	16	14.6	12.7	(−)15	12.6	(−)23	5.9 ± 2.0 [65]
Finger-Tapping Test (n)	35	32.3	36.6	9	42.3	26	21	19.6	33	62	34.6	70	48.3 ± 5.0 [66]
Finger-to-Nose Test (n)	24.6	25.3	27.3	9	28.3	12.5	*	*	*	-	*	-	-
Grip Strength (kg)	#	#	#	-	#	-	4.3	4.6	6.3	17	7.6	18	38 ± 8.0 [67]

+ Could not perform and hence were given the maximum value of 120 s; * Could not perform finger-to-nose test because of shoulder pain; # JAMAR handgrip force assessment was not evaluated for Participant 1, ** Mean ± SD values are reported for all age-related norms; For the nine-hole peg test and Jebsen hand function test, a lower value indicates a better performance. Abbreviations: Base: Baseline; Pre: pre-training; Post: post-training; F-up: Follow up, %: percentage change from the average of baseline and pre-training blocks (week 0, week 3).

## Data Availability

The quantitative data regarding the clinical and neurophysiological assessments can be provided upon request.

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
