# Peer review of "Neurophysiological Changes Induced by Music-Supported Therapy for Recovering Upper Extremity Function after Stroke: A Case Series"

_brainsci, 2021, doi:10.3390/brainsci11050666_

Round 1

Reviewer 1 Report

Undoubtedly, MST follows the best practice principles of stroke rehabilitation and has been proven to instigate meaningful enhancements in motor recovery post-stroke. In this article, Authors evaluated changes in connectivity within the auditory-motor network and changes in upper extremity function following a three weeks intensive piano training in two stroke survivors presenting different levels of motor impairment. 

My comments to the article are as follows:

- As part of the Introduction, I propose to extend the background by referring to the issue of the impact of sounds on the level of stress measured, among others by measuring the EEG signal. For example, you can refer to the work: The Impact of Different Sounds on Stress Level in the Context of EEG, Cardiac Measures and Subjective Stress Level: A Pilot Study, Brain Sciences from 2020. Additionally, it will have an impact on updating the bibliography which, despite the fact that it is extensive, often contains articles older than the last 5 years.

- On Fig. 1 the units are not entirely clear to me. Please explain them. You have [Hz] -> [Hz] ?

- Please, clearly indicate in which software the models on Fig. 2 were created.

- Conlusions need to be expanded, they are too short. They should be expanded to include future plans for this research.

Author Response

Response

Reviewer 1

Undoubtedly, MST follows the best practice principles of stroke rehabilitation and has been proven to instigate meaningful enhancements in motor recovery post-stroke. In this article, Authors evaluated changes in connectivity within the auditory-motor network and changes in upper extremity function following a three weeks intensive piano training in two stroke survivors presenting different levels of motor impairment.

My comments to the article are as follows:

Comment: As part of the Introduction, I propose to extend the background by referring to the issue of the impact of sounds on the level of stress measured, among others by measuring the EEG signal. For example, you can refer to the work: The Impact of Different Sounds on Stress Level in the Context of EEG, Cardiac Measures and Subjective Stress Level: A Pilot Study, Brain Sciences from 2020. Additionally, it will have an impact on updating the bibliography which, despite the fact that it is extensive, often contains articles older than the last 5 years.

Response: We thank the reviewer for this comment. Indeed, the article referred by you explains the strong influence of sounds on the stress levels. We have now mentioned this in our introduction.

The changes in introduction are mentioned at:

Page 2, Line 58-59: “Previous studies including the ones from our laboratory [15,16], indicate that music-supported therapy (MST) can not only meet but extend beyond these imperatives to yield improvements in motor, cognitive functions [17-23], and stress reduction [24].”

Comment: On Fig. 1 the units are not entirely clear to me. Please explain them. You have [Hz] -> [Hz] ?

Response: The SI units are signal unit per square root Hz. We have now amended this in the figure for clarity.

Comment: Please, clearly indicate in which software the models on Fig. 2 were created.

Response: Brainstorm software (https://neuroimage.usc.edu/brainstorm) was used to create these figures. We have now mentioned this in the figure caption of Figure 2 and also on Page 5, Line 195.

Comment: Conclusions need to be expanded; they are too short. They should be expanded to include future plans for this research.

Response: We agree and have now mentioned that the findings of this case series would be used to develop further clinical trials in our laboratory. The changes are mentioned on

Page 13, Line 435: “We look forward to these findings contributing to delineating a path for future clinical trials in our laboratory. Additionally, in future trials we intend to evaluate the global mapping of possible brain regions showing variations of CMC with motor functions. These changes will help us identify which brain nodes in motor circuits interact with the effectors (hand muscles) to support the recovery of paretic hand after MST.”

Reviewer 2 Report

Review for “Neurophysiological changes induced by music-supported therapy for recovering upper-extremity function after stroke: A case 3 series”

 In this study 2 stroke patients underwent training in music supported therapy that required the patients to play piano and receive feedback on what they heard. The training included supervised and at home sessions. Individuals were examined on behavioral progress (grip strength) and MEG activity twice before intervention and twice after intervention.  Both patients showed significant behavioral progress following intervention. Also, both patients showed changes of activity in several regions including auditory cortex and motor areas. Most importantly, using source analysis they showed that connectivity between motor and auditory areas are enhanced following training.

This is an interesting study, both to scientists and public.  It is disappointing that the study did not include all 7 participants, due to COVID. Nonetheless, I believe this is still an impactful work to the field.

I have major and minor concerns:

Major:

1-    The study did not include patients who did not undergo training. Due to the lack of control patients we cannot be sure whether changes are attributed to training or normal recovery. The authors need to at least include this as a limitation of the study.

2-    While I appreciate the supplementary data of current density distribution, I would also appreciate before and after topographies (MEG scalp distribution) of the beta and alpha band; the same scalp distribution that the sources analysis was based on.

3-    Connectivity due to source analysis can be problematic, as slight changes in orientation, or depth (especially in MEG) can result in the change of solutions at all tested sources. Do the dipole orientations stay constant in orientation and depth (I assume)? If so, please state this fact.

Minor:

Please define the full regions names at the biggining (M1, PMd, PMv, SMA, IFG, and 225 the AC. SMA, PMv, PMd).

Author Response

Reviewer 2

Review for “Neurophysiological changes induced by music-supported therapy for recovering upper-extremity function after stroke: A case 3 series”

 In this study 2 stroke patients underwent training in music supported therapy that required the patients to play piano and receive feedback on what they heard. The training included supervised and at home sessions. Individuals were examined on behavioral progress (grip strength) and MEG activity twice before intervention and twice after intervention.  Both patients showed significant behavioral progress following intervention. Also, both patients showed changes of activity in several regions including auditory cortex and motor areas. Most importantly, using source analysis they showed that connectivity between motor and auditory areas are enhanced following training.

This is an interesting study, both to scientists and public.  It is disappointing that the study did not include all 7 participants, due to COVID. Nonetheless, I believe this is still an impactful work to the field.

I have major and minor concerns:

Major:

Comment: The study did not include patients who did not undergo training. Due to the lack of control patients we cannot be sure whether changes are attributed to training or normal recovery. The authors need to at least include this as a limitation of the study.

Response: We thank the reviewer for this comment. As mentioned in the article, we employed a single subject design referred to as ‘multiple pre- multiple post- sequential design’ in which the participants, who were in the chronic phase of stroke (> 15 months), did not undergo any sort of training during the initial 3 weeks and thus served as their own controls. While such design does not allow concluding on comparative effectiveness (or to dissociate the placebo effect from actual intervention effect), it is appropriate given the purpose of the study which was to examine the neurophysiological changes induced by MST and how these changes are related to the clinical profile of the participants. Also, the purpose of the multiple baseline evaluations prior to the intervention is to demonstrate that participants are stable and that no more natural recovery is taking place. Accordingly, we noted no changes in the performance of participants on clinical tests between the two baseline evaluations preceding the intervention (see Table 2). It is thus very unlikely that changes observed post-intervention are due to natural recovery. Nevertheless, we understand and acknowledge the limitations inherent to such study design and have thus amended the limitation section of our article. The changes read as follows:

Page 13. Line 423-428: “Fourthly, this study used a single subject design referred to as ‘multiple pre- multiple post- sequential design’, instead of using a placebo group. While such design does not allow concluding on comparative effectiveness (or to dissociate the placebo effect from actual intervention effect), it is appropriate given the purpose of the study which was to examine the neurophysiological changes induced by MST and how these changes are related to the clinical profile of the participants”  

Comment: While I appreciate the supplementary data of current density distribution, I would also appreciate before and after topographies (MEG scalp distribution) of the beta and alpha band; the same scalp distribution that the sources analysis was based on.

Response: As per your suggestion, we have now included the MEG scalp topographies in the Supplementary file of the manuscript. The new figures are supplementary figures 7-9.

Comment: Connectivity due to source analysis can be problematic, as slight changes in orientation, or depth (especially in MEG) can result in the change of solutions at all tested sources. Do the dipole orientations stay constant in orientation and depth (I assume)? If so, please state this fact.

Response: We thank the reviewer for this critical comment. Indeed, the dipole orientations were constant for orientation. Specifically, the orientation was constrained perpendicularly with respect to the cortical surface. Moreover, we used depth weighting with an order of 0.5 and maximal amount of 10. This has now been mentioned in the manuscript:

Page 5, Line 206-208: The dipole orientations were kept constant as they were constrained perpendicularly with respect to the cortical surface. Moreover, we used depth weighting with an order of 0.5 and maximal amount of 10.

Minor:

Comment: Please define the full regions names at the beginning (M1, PMd, PMv, SMA, IFG, and 225 the AC. SMA, PMv, PMd).

Response: The full names of the ROIs have now been stated at the beginning of Page: 1, Line 44-45.